# Preventability of unplanned readmissions within 30 days of discharge. A cross-sectional, single-center study

Albertine M. B. van der Does[1ᵒ], Eva L. Kneepkens[1ᵒ], Elien B. Uitvlugt[1ᵒ], Sanne L. Jansen[2‡], Louise Schilder[3‡], George Tokmaji[4‡], Sofieke C. Wijers[3‡], Marijn Radersma[5‡], J. Nina M. Heijnen[6‡], Paul F. A. Teunissen[3‡], Pim B. J. E. Hulshof[3‡], Geke M. Overvliet[7‡], Carl E. H. Siegert[3ᵒ], Fatma Karapinar-Çarkit[1ᵒ]*

1 Department of Clinical Pharmacy, OLVG, Amsterdam, The Netherlands, 2 Department of Surgery, OLVG, Amsterdam, The Netherlands, 3 Department of Internal medicine, OLVG, Amsterdam, The Netherlands, 4 Department of Cardiology, OLVG, Amsterdam, The Netherlands, 5 Department of Gastroenterology, OLVG, Amsterdam, The Netherlands, 6 Department of Neurology, OLVG, Amsterdam, The Netherlands, 7 Department of Psychiatry, OLVG, Amsterdam, The Netherlands

ᵒ These authors contributed equally to this work.
‡ These authors also contributed equally to this work.
* f.karapinar@olvg.nl

**Data Availability Statement:** All relevant data are within the paper and its Supporting Information files.

## Abstract

### Objectives

To identify the preventability, determinants and causes of unplanned hospital readmissions within 30 days of discharge using a multidisciplinary approach and including patients' perspectives.

### Design

A prospective cross-sectional single-center study.

### Setting

Urban teaching hospital in Amsterdam, the Netherlands.

### Participants

430 patients were included. Inclusion criteria were: age ≥ 18 years, discharged from one of seven participating clinical departments and an unplanned readmission within 30 days.

### Methods

Residents from the participating departments individually assessed whether the readmission was caused by healthcare, the preventability and possible causes of readmissions using a tool. Thereafter, the preventability of the cases was discussed in a multidisciplinary meeting with residents of all participating departments and clinical pharmacists. The primary outcome was the proportion of readmissions that were potentially preventable. Secondary outcomes were the determinants for a readmission, causes for preventable readmissions,

**Funding:** FK received a grant from the inhouse OLVG innovation fund (2015). The innovation fund has no website/URL and issue no grant numbers. Calls for proposals are communicated twice a year through the inhouse hospital newsletter. The funders had no role in study design, data collection and analysis, decision to publish, or preparation of the manuscript.

**Competing interests:** The authors have declared that no competing interests exist.

the change in the final decision on preventability after the multidisciplinary meeting and the value of patient interviews in assessing preventability. Differences in characteristics of potentially preventable readmissions (PPRs) and non-PPRs were analyzed using multivariable logistic regression.

## Results

Of 430 readmissions, 56 (13%) were assessed as PPRs. Age was significantly associated with a PPR (adjusted OR: 2.42; 95%, CI 1.23–4.74; $p$ = 0.01). The main causes for PPRs were diagnostic (30%), medication (27%) and management problems (27%). During the multidisciplinary meeting, the final decision on preventability changed in 11% of the cases. When a patient interview was available, it was used as a source of information to assess preventability in 26% of readmissions. In 7% of cases, the patient interview was mentioned as the most important source.

## Conclusion and implications

13% of readmissions were potentially preventable with diagnostic, medication or management problems being main causes. A multidisciplinary review approach and including the patient's perspective could contribute to a better understanding of the complexity of readmissions and possible improvements.

## Introduction

Unplanned readmissions are a stressful and disappointing event for both patients and healthcare professionals. Also, unplanned readmissions within a month after discharge can be used as a parameter of quality of care [1,2]. Readmission rates above the benchmark have even been used as financial penalties for healthcare institutions [3]. Using readmission rates as a quality parameter illustrates a belief that readmissions are avoidable and a sign of insufficient care, and this assumption is being questioned [4,5,6]. Researchers argue that not all readmissions are preventable and that the proportion of potentially preventable readmissions (PPRs) would be a better parameter of quality of care, rather than the total number of readmissions [6]. Focusing on PPRs would also identify areas of improvement [4]. Measuring preventability, however, poses some challenges since it lacks a clear definition and objective measuring tool. This causes confusion in methodology and comparability of study results, possibly explaining the wide variety in reported preventable readmissions [5–10]. In a review on hospital admissions considered avoidable, a preventability proportion between 5% and 79% was reported [7].

In most studies, a preventability assessment is performed by one or more attending physicians [7]. Their assessment may be limited to looking for causes mainly within their own department and clinical specialty. Although still a subject of debate, a multidisciplinary approach may help to capture the patient's complexity in terms of comorbidities, polypharmacy, nursing care, or social and psychological needs [7,9,11]. Within this multidisciplinary approach, the preferred method to examine harm due to care is the patient's chart review [2], as opposed to only using discrete data obtained from hospital systems. By using the patient's chart as a source, all relevant information is accessible. Additional information from, for example, nurse or physician summaries or from the patient himself may play an important role in assessing preventability [7,9,11].

Most studies on the causes and preventability of readmissions are performed in the United States and the United Kingdom [7], while availability of preventability rates for other European countries is limited.

Therefore, the aim of the current study is to assess the preventability, determinants and causes, of unplanned hospital readmissions within thirty days of hospital discharge, using a multidisciplinary approach and including patients' perspectives. Also, potential preventive actions were assessed.

## Research design and methods

### Study design and setting

A cross-sectional single-center observational study was conducted from 15 July 2016 until 30 April 2017. Patients were included after being admitted to a teaching hospital in Amsterdam, the Netherlands. Inclusion criteria were: unplanned readmissions of adult patients ($\geq$ 18 years) within thirty days after discharge from an earlier admission (index admission = IA) from one of the participating departments: cardiology, gastroenterology, internal medicine, neurology, psychiatry, pulmonology and general surgery. Participating departments were selected based on the highest number of unplanned readmissions during previous years. Only a patient's first readmission was included.

Exclusion criteria were: patients who were transferred to another hospital during IA and patients who left the hospital against medical advice during IA. Furthermore, a readmission was excluded if it was deemed unrelated to the IA.

The study was approved by the Advisory Committee Scientific Research of OLVG hospital (Advies Commissie Wetenschappelijk Onderwijs). Informed consent was obtained from patients for the interview and to contact the community pharmacist or general practitioner for additional information (e.g., on medication adherence).

### Study process and data collection

**Screening of readmissions.**  A list of unplanned readmissions was generated within the hospital information system showing all readmitted patients within thirty days of discharge. This list was manually screened by the study coordinator, who was a medical doctor, for inclusion, during week days. The study coordinator initially assessed if the readmission was related to the IA. This was double checked by the resident of the department of the IA. If the two assessments were not in agreement, the case was discussed in a multidisciplinary meeting (see below).

**Assessments by residents.**  After inclusion criteria were met, the resident of the discharging department during the IA was asked to assess the IA and readmission using a review tool. The purpose of the resident review tool was to structure the assessment process of the residents, making the scoring process more uniform across departments and to facilitate possible discussion. The review tool consisted of a semi-structured questionnaire. Admission diagnoses and the presence of contributing factors to the readmission were noted (S1 and S2 Tables). Then, causation was scored using a six-point scale (S3 Table). Causation was defined as the extent to which the provided care during IA, and the subsequent outpatient follow-up care provided by the hospital, caused the readmission. A score $\geq$ 4 was defined as a causal readmission, on a scoring scale of 1 to 6 [12,13]. Subsequently, preventability was assessed for readmissions with a causation score $\geq$ 4, on a scoring scale of 1 to 6 (S3 Table). A readmission was considered preventable if certain action(s) (not) taken during IA or subsequent outpatient follow-up care provided by the hospital could have prevented the readmission, taking into consideration relevant current guidelines.

If a readmission received a preventability score of four or higher, causes and possible preventive actions were noted in free text fields. The causes were based on the following

categories: surgery, procedural, nosocomial infection, medication, diagnostic, management or system error (S4 Table).

For determination of causality and preventability, the residents received the discharge letter of the IA, the admission notes of the readmission and a transcript of the patient interview, including a summary. For additional information, the residents could consult the patient file in the hospital information system. In the resident review tool, residents noted which information was used to assess preventability (e.g., patient interview, nurse summary, discharge letter) and which information was crucial.

**Assessments by multidisciplinary meetings.**   The resident-reviews were evaluated by the research coordinator and a clinical pharmacist. All reviews that scored $\geq 4$ on preventability or that seemed unclear were selected to be discussed during a multidisciplinary meeting attended by all residents of the participating departments. During these meetings, consensus on the preventability was assessed, resulting in the definitive preventability score. Again, a score of $\geq 4$ represented the assessment by the group, that a readmission was potentially preventable with an estimated chance of $> 50\%$ (S3 Table).

Secondly, during these meetings the reasons why the readmission was deemed preventable, the causes of the preventable admission and possible interventions that could have prevented the readmission were discussed. To assess the additional value of the multidisciplinary meeting in assessing preventability, the research coordinator documented how often the preventability score changed due to the multidisciplinary meeting.

**Residents' training.**   All participating departments and the hospital pharmacy supported the study with at least one resident to complete the reviews.

All residents received a group training prior to the start of the study regarding the review process and tool. A manual was available to guarantee consistency in the review process. The resident review tool was developed based on previous studies and expert opinion [12–14].

In accordance with previous studies [15], all residents could verify their conclusions with a senior professional. The first month of this study was a pilot phase to evaluate the feasibility and reliability of the screening, interview and review process. Based on the outcome of the pilot, several minor adaptations were made. The readmissions that were included during the pilot phase, were included in the final analysis.

**Patients' interview.**   Upon inclusion, the patient or the caregiver was interviewed during admission. If the patient was already discharged, patients were contacted by phone. A maximum of three telephonic attempts were made.

The patient perspective was explored by a trained medical student, using a semi-structured interview guide. The structured patient interview guide was developed based on the available literature and expert opinion [16–21]. The topics addressed were: patient consultation with general practitioner, adherence to medical advice (e.g., medication, life style and dietary restrictions), presence of social support and patient's perspectives on preventability of the readmission. Additionally, information on socio-demographic characteristics and health literacy (scoring scale 0–4,) using the "Set of Brief Screening Questions" [19,21], was gathered. Readiness for discharge and the self-perceived health status were documented using the B-prepared questionnaire [20]. To assess the value of the patient interview, the number of times that the patient interview was used as a source to assess preventability was extracted from the resident review tool.

## Outcomes

The primary outcome was the proportion of readmissions that was assessed as potentially preventable (causality score $\geq 4$ and preventability score $\geq 4$).

The secondary outcome variables considered the determinants for a PPR compared to a non-PPR and causes of PPRs. The additional value of multidisciplinary meetings and the patient interview in assessing preventability was evaluated. Finally, potential preventive actions were assessed.

### Statistical analysis

The Statistical Package for the Social Sciences (SPSS) version 21.0 (IBM Analytics) was used. Categorical variables are reported as frequencies. Normally or non-normally distributed continuous variables are reported as the mean with the standard deviation (SD) or median with the interquartile range (IQR), respectively, unless stated otherwise.

Multivariable logistic regression analysis was used to compare PPRs and non-PPRs and assess determinants, adjusting for possible confounding. The determinants assessed were socio-demographic data and admission characteristics. A manual stepwise forward logistic model was used. Possible confounders ($p < 0.2$) were entered consecutively into the model. When the β-coefficient changed $\geq 10\%$, the contribution was considered relevant and the confounder remained in the model. Crude and adjusted odds ratio's (ORs) with 95% confidence intervals (95% CIs) and $p$-values were calculated.

For the preventive actions, a qualitative data exploration was used, independently assessed by EK and EU. Themes were identified to categorize these interventions.

## Results

A total of 646 readmissions were screened. Of these readmissions, 94 (15%) were considered to be unrelated to the IA. This resulted in the assessment of 552 (85%) related readmissions for 430 unique patients. Table 1 shows the patient characteristics and admission characteristics. Included patients had a mean age of 63 years (SD 17.6) and gender was equally divided between PPRs and non-PPRs.

### Causality and preventability scoring

Of 430 first readmissions, 201 (47%) had a causation score of $\geq 4$. Fifty-six readmissions were subsequently considered to be preventable during multidisciplinary meetings (13% of the included 430 readmissions in total).

### Determinants for PPR

Patients with a PPR were significantly older than patients with a non-PPR (62 years versus 69 years, $p = 0.011$). Older age (i.e., older than 65 years) was significantly associated with a PPR (unadjusted OR 1.9 (95% CI 1.0–3.6; $p = 0.049$); adjusted OR 2.42 (95% CI 1.23–4.74; $p = 0.01$). All other characteristics in Table 1 showed no significant difference between PPRs and non-PPRs.

### Causes of potential preventable readmissions

Fig 1 shows the causes of potential preventable readmissions. The most reported causes were diagnostic (30%), medication (27%) and management (27%) problems.

### Multidisciplinary meetings

Thirteen multidisciplinary meetings were held to discuss 106 (25%) of 430 readmissions. The group discussion resulted in an increase of the preventability in five cases (5%) and a non-preventable conclusion in seven (7%). Thus, in 11% of the cases, the multidisciplinary meeting

**Table 1. Patient and admission characteristics (n = 430).**

| Characteristics | Total population n = 430 | Non-preventable readmissions n = 374 (87%) | Preventable readmissions[a] n = 56 (13%) |
|---|---|---|---|
| *Patient characteristics* | | | |
| Age in years, mean (SD) | 62.9 (17.6) | 62 (17.5) | 68.5 (17.3) |
| Male, n (%) | 211 (49.1) | 180 (48.1) | 31 (55.4) |
| Language barrier present, n (%) | 88 (20.5) | 79 (21.1) | 9 (16.1) |
| Living alone, n (%) | 203 (47.2) | 176 (47.1) | 27 (48.2) |
| Discharged to home, n (%) | 376 (87.4) | 329 (88) | 47 (83.9) |
| ≥ 2 previous hospital admissions, n (%) | 60 (14) | 55 (14.7) | 5 (8.9) |
| *Index admission characteristics* | | | |
| Unplanned admission, n (%) | 334 (77.7) | 291 (77.8) | 43 (76.8) |
| Duration of stay in days (range) | 4 (2–9.3) | 4.5 (2–9) | 4 (1.3–11.8) |
| ≥ 3 medication changes, n (%) | 184 (42.8) | 157 (42) | 27 (48.2) |
| ≥ 5 medicines at discharge, n (%) | 312 (72.6) | 270 (72.2) | 42 (75) |
| Medication reconciliation at discharge, n (%) | 201 (46.7) | 179 (47.9) | 22 (39.3) |
| Discharge on Saturday or Sunday, n (%) | 63 (14.7) | 54 (14.4) | 9 (16.1) |
| Discharge letter sent ≤ 2 days, n (%) | 120 (27.9) | 103 (27.5) | 17 (30.4) |
| Planned post-discharge outpatient visit, n (%) | 367 (85.3) | 323 (86.4) | 44 (78.6) |
| Time until readmission, days (range) | 9 (4–17) | 10 (4–18) | 7 (2–13) |

[a] Potential preventable readmissions as assessed in the multidisciplinary meetings

resulted in a change in the final preventability conclusion. In another twenty cases (19%), a change in the score was made but without consequences for the final preventability conclusion.

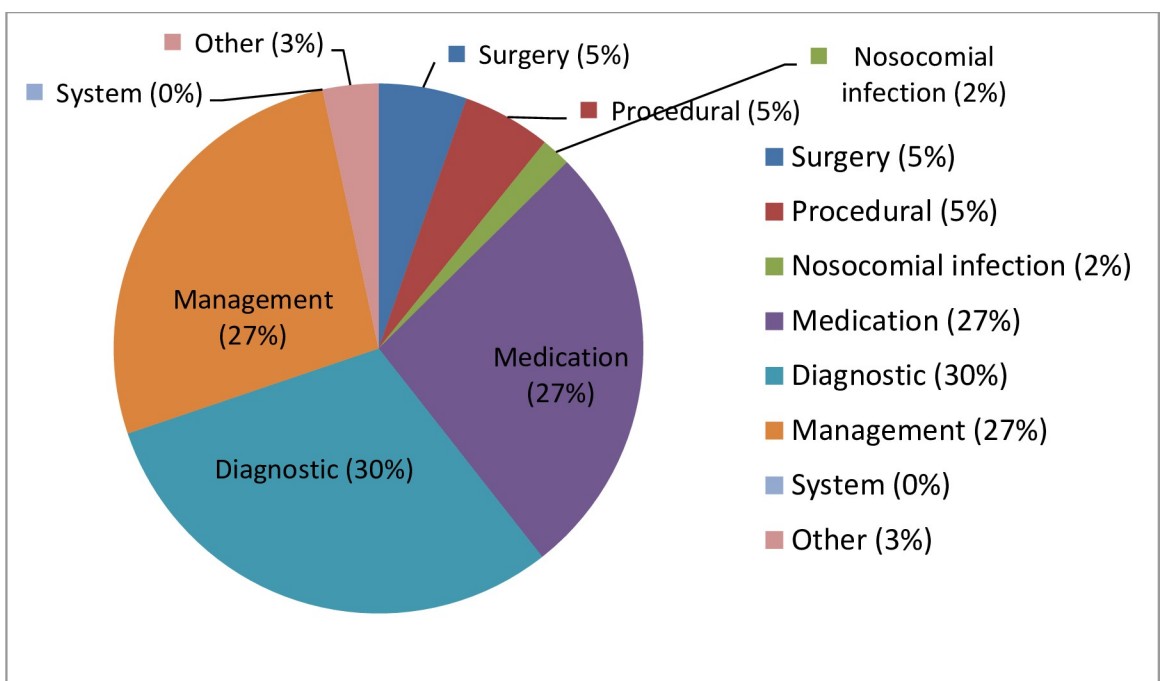

**Fig 1. Causes of potentially preventable readmissions.** Causes of preventable readmissions as decided in the multidisciplinary meetings (n = 56).

**Table 2. Results of patient and/or carer interviews (n = 227).** The results are divided by non-preventable readmissions (non-PPRs) versus potentially preventable readmissions (PPRs) based on the multidisciplinary group discussion of caregivers.

| | Population with a complete interview n = 227 | Non-PPRs n = 196 (86.3%) | PPRs n = 31 (13.7%) |
|---|---|---|---|
| Patients interviewed, n (%) | 200 (88.1) | 174 (88.8) | 26 (83.9) |
| Dutch nationality, n (%) | 141 (62.1) | 122 (62.2) | 19 (61.3) |
| Low education level, n (%) | 41 (18.1) | 35 (17.9) | 6 (19.4) |
| Social support available, n (%) | 189 (83.3) | 162 (82.7) | 27 (87.1) |
| Poor self-experienced health status, n (%) | 86 (37.9) | 71 (36.2) | 15 (48.4) |
| Inadequate health literacy[a], n (%) | 51 (22.5) | 44 (22.4) | 7 (22.6) |
| B-Prepared, mean score[b] ±SD | 16.7 ± 4.2 | 16.6 ± 4.3 | 16.9 ± 3.6 |
| Patients reported dietary and lifestyle advice, n (%) | 72 (31.7) | 67 (34.2) | 5 (16.7) |
| Patients reported adherence to dietary and lifestyle restrictions | 49 (68.1) | 45 (67.2) | 4 (80.0) |
| Visited general practitioner[c], n (%) | 104 (45.8) | 86 (43.9) | 18 (58.1) |
| Discharged prematurely, n (%) | 84 (37) | 72 (36.7) | 12 (38.7) |
| Preventive actions possible, n (%) | 106 (46.7) | 92 (46.9) | 14 (45.2) |
| Expected readmission, n (%) | 54 (23.8) | 53 (27) | 1 (3.2) |

[a]Inadequate health literacy based on score that ranges from 0–4. "Low" was set at a score of 2 or lower. SBSQ questionnaire used[19,21].

[b]B-Prepared score that ranges from 0–22, a higher score represents a higher level of perceived preparedness[20].

[c]The patient visited a general practitioner between the index admission and readmission.

### Patient interviews

Two-hundred and twenty-seven interviews (53%) were conducted: 200 with the patient and 27 with a caregiver. An interview was not available for 203 (47%) patients. Causes were cognitive and/or physical problems (n = 34), severe illness (n = 23), a language barrier (n = 29), death (n = 13), unwillingness to participate (n = 39) or failed attempts to contact the patient (n = 50), as well as miscellaneous causes (n = 15).

Table 2 shows the results of the patient interviews, divided by non-PPRs versus PPRs (based on the definitive outcome determined during the multidisciplinary meeting). Some tentative results were that patients with a PPR had consulted the general practitioner more often, expected less often that they would be readmitted, and more often had a poor self-experienced health status.

Agreement on preventability as considered by the patient (or their carer) and as determined by the consensus score in the meeting by the health care professionals occurred for 19 of 56 (34%) of the cases.

The patient interview was mentioned 15 times as a crucial source to assess preventability (7% of 227 interviews). The patient interview was also mentioned 60 times as a secondary source and more often in PPRs (PPRs vs non-PPRs: 36% vs 11%, $p$ = 0.07).

For information on all sources used by the residents, see S1 Data.

### Preventive actions

Table 3 shows the synopsis of themes based on the free text fields on the suggested preventive actions of 56 PPRs. Over half of readmissions were deemed preventable by more comprehensive diagnostic assessment and closer monitoring of and acting on insufficient responses to treatment.

### Discussion

This study identified that 13% of readmissions are potentially preventable. Age was significantly associated with a PPR (adjusted OR: 2.42; 95% CI 1.23–4.74; $p$ = 0.01). The main causes

**Table 3. Possible interventions to prevent readmissions according to residents (n = 56).**

| Intervention categories | Number (%) |
|---|---|
| 1. Broader differential diagnosis and/or additional investigation needed | 12 (21%) |
| 2. More strict evaluation of treatment outcome (and if applicable, action upon treatment outcome) | 10 (18%) |
| 3. A more adequate work-up/assessment of symptoms, complaints and/or care need. | 9 (16%) |
| 4. Medication policy (e.g., other medicine(s), slower or quicker dose changes) | 7 (13%) |
| 5. Consultation of another specialized care provider and/or another hospital department | 7 (13%) |
| 6. Longer length of stay/observation time at index admission | 6 (11%) |
| 7. Better communication with other care providers | 2 (4%) |
| 8. Better patient education | 2 (4%) |
| 9. Unclassifiable | 1 (2%) |

for PPRs were diagnostic, medication and management problems. A multidisciplinary review approach changed the final conclusion on preventability in 11% of cases. The patient interview was crucial for drawing a conclusion in 7% of readmissions and used as a source to inform the review in 26% of readmissions.

The percentage of preventable readmissions, 13%, was lower than expected [7]. However, most previous studies were performed in the USA. The causes for this difference may be due to patient mix, the definition of preventability and the context of the study. Our findings are consistent with a recent European study, which was partly conducted in The Netherlands and found 14% of readmissions to be preventable [22].

Readmitted patients were significantly older than non-readmitted patients. No other determinants were found in this study. Other studies reported different determinants for preventable readmissions, which could be due to differing patient mix, sample size and how preventability was assessed in the different studies [23–25]. Further research into the determinants is needed.

Three main causes for PPRs were found: diagnostic (e.g., misdiagnosis or delayed diagnosis), medication (e.g., incorrect prescription or use) and management problems (e.g., inadequate discharge planning or integrated care/transition of care issues). These finding were largely comparable with the results of van Walraven et al [14]. The multidisciplinary approach of using residents from seven departments and pharmacists might explain the main causes that were found in this study; a multidisciplinary approach is likely to contribute to a broader view on the patient's symptoms and care needs. Interdepartmental, integrated care issues and medication errors were recognized by the different members of the team. Previous studies using a multidisciplinary approach are limited [15,23,26,27] and, to our knowledge, none of them specifically described the added value of this approach. The multidisciplinary meeting resulted in a modification of the final conclusion on preventability in 1 in 9 cases. These findings underline the value of the multidisciplinary approach; it is helpful to more comprehensively assess readmissions, considering the complexity of most patients, especially the elderly, and the number of patients readmitted within thirty days to other departments [24].

The patient interview was crucial for residents in assessing preventability in 1 in 14 cases. We did not systematically assess which specific information in the patient interview was of crucial value for the preventability assessment. To our knowledge, only one study in pediatric patients systematically assessed the value of the patient's perspective [3]. Future studies should also take this into account. Also, the perspectives of primary care providers should be included [4,9]. Currently, readmissions are used to monitor quality of *hospital* care; therefore, preventability is often assessed from a hospital's perspective. However, patients and primary care providers report on other issues regarding readmissions as compared to hospital care providers [28].

A strength of this study is the inclusion of a broad range of departments, including a multi-disciplinary approach and the patient's perspective. However, limitations also need to be discussed. Firstly, although we included patients prospectively, the retrospective assessment of preventability could result in hindsight and recall bias. However, the time between inclusion and the review of cases was short, and the influence of hindsight bias was discussed during the multidisciplinary meetings. Secondly, not all readmission cases were discussed during the multidisciplinary meetings in order to increase the practical feasibility of this research. Inclusion was determined by subjective assessment by a member of the discharging department. This could lead to non-inclusion of patients who might have had a PPR, thus preventable readmission cases could therefore have been underestimated.

However, the coordinating researcher and clinical pharmacist did additional checks and included readmission cases for the multidisciplinary meeting if questions arose. Thirdly, the coordinating researcher screened reviews for inconsistencies and protocol compliance. However, the interrater agreement was not calculated—instead regular multidisciplinary meetings were conducted to obtain consensus—a senior consultant could be asked for input and a group training was provided, an approach consistent with Auerbach et al [29]. Previous studies have also worked with residents as reviewers [30,31].

## Conclusion and implications

In conclusion, one in eight readmissions were regarded potentially preventable. Patients with PPRs were older. Diagnostic, medication or management problems were major causes of PPRs. A multidisciplinary review approach and including the patient's perspective could contribute to a better understanding of the complexity of readmissions and possible improvements.

## Supporting information

**S1 Table. Definition of each contributing factor.**
(DOCX)

**S2 Table. Factors contributing to PPRs vs non-PPRs.** The table represents the number (%) of contributing factors for three groups: total population, non-preventable readmissions and possible preventable readmissions.
(DOCX)

**S3 Table. Causation and preventability scoring tools.** Reviewers used a six-point ordinal scale to rate whether the readmission was causal (readmission due to medical care during the index admission) and whether the readmission could have been prevented.
(DOCX)

**S4 Table. Cause classification.** Cause categories for readmissions and examples for each category.
(DOCX)

**S1 File. Sources used by residents for the causation and preventability assessment.**
(DOCX)

**S2 File.**
(DOCX)

**S3 File.**
(DOCX)

**S1 Data.**
(SAV)

**S1 Fig.**
(DOCX)

**S2 Fig.**
(DOCX)

## Acknowledgments

The authors are grateful to all patients and departments that participated in this research and to the department of Public and Occupational health of the EMGO+ Institute for Health and Care Research, Amsterdam, the Netherlands for assisting in the training of residents in assessing preventability.

## Author Contributions

**Conceptualization:** Eva L. Kneepkens, Elien B. Uitvlugt, Carl E. H. Siegert, Fatma Karapinar-Çarkit.

**Data curation:** Albertine M. B. van der Does, Eva L. Kneepkens, Elien B. Uitvlugt, Fatma Karapinar-Çarkit.

**Formal analysis:** Albertine M. B. van der Does, Eva L. Kneepkens, Elien B. Uitvlugt.

**Funding acquisition:** Carl E. H. Siegert, Fatma Karapinar-Çarkit.

**Investigation:** Eva L. Kneepkens, Elien B. Uitvlugt, Sanne L. Jansen, Louise Schilder, George Tokmaji, Sofieke C. Wijers, Marijn Radersma, J. Nina M. Heijnen, Paul F. A. Teunissen, Pim B. J. E. Hulshof, Geke M. Overvliet, Carl E. H. Siegert, Fatma Karapinar-Çarkit.

**Methodology:** Eva L. Kneepkens, Elien B. Uitvlugt, Carl E. H. Siegert, Fatma Karapinar-Çarkit.

**Project administration:** Eva L. Kneepkens, Fatma Karapinar-Çarkit.

**Resources:** Eva L. Kneepkens, Fatma Karapinar-Çarkit.

**Supervision:** Carl E. H. Siegert, Fatma Karapinar-Çarkit.

**Validation:** Albertine M. B. van der Does, Fatma Karapinar-Çarkit.

**Visualization:** Albertine M. B. van der Does, Fatma Karapinar-Çarkit.

**Writing – original draft:** Albertine M. B. van der Does, Eva L. Kneepkens.

**Writing – review & editing:** Albertine M. B. van der Does, Elien B. Uitvlugt, Sanne L. Jansen, Louise Schilder, George Tokmaji, Sofieke C. Wijers, Marijn Radersma, J. Nina M. Heijnen, Paul F. A. Teunissen, Pim B. J. E. Hulshof, Geke M. Overvliet, Carl E. H. Siegert, Fatma Karapinar-Çarkit.

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
