## [Decision Letter · Decision Letter 0]

7 Jan 2020

PONE-D-19-31179

Preventability of unplanned readmissions within 30 days of discharge: A cross-sectional, single-centre study.

PLOS ONE

Dear Drs van der Does,

Thank you for submitting your manuscript to PLOS ONE. After careful consideration, we feel that it has merit but does not fully meet PLOS ONE’s publication criteria as it currently stands. Therefore, we invite you to submit a revised version of the manuscript that addresses the points raised during the review process.

We would appreciate receiving your revised manuscript by Feb 21 2020 11:59PM. To enhance the reproducibility of your results, we recommend that if applicable you deposit your laboratory protocols in protocols.io, where a protocol can be assigned its own identifier (DOI) such that it can be cited independently in the future. For instructions see: http://journals.plos.org/plosone/s/submission-guidelines#loc-laboratory-protocols

We look forward to receiving your revised manuscript.

Kind regards,

Peter Dziegielewski, MD, FRCSC

Academic Editor

PLOS ONE

Journal Requirements:

2. Please include additional information regarding the interview guide used in the study and ensure that you have provided sufficient details that others could replicate the analyses. For instance, if you developed a guide as part of this study and it is not under a copyright more restrictive than CC-BY, please include a copy, in both the original language and English, as Supporting Information."

3. Thank you for including your ethics statement:

"The study was approved by the scientific board of the hospital (ACWO) with approval number: 16-028. Informed consent was obtained from patients for the interview and to contact the community pharmacist or general practitioner for additional information (e.g., on medication adherence)."

b) Once you have amended this statement in the Methods section of the manuscript, please add the same text to the “Ethics Statement” field of the submission form (via “Edit Submission”).

Reviewers' comments:

Reviewer's Responses to Questions

**Comments to the Author**

1. Is the manuscript technically sound, and do the data support the conclusions?

Reviewer #1: Yes

Reviewer #2: Yes

2. Has the statistical analysis been performed appropriately and rigorously? 

Reviewer #1: Yes

Reviewer #2: Yes

3. Have the authors made all data underlying the findings in their manuscript fully available?

Reviewer #1: Yes

Reviewer #2: Yes

4. Is the manuscript presented in an intelligible fashion and written in standard English?

Reviewer #1: Yes

Reviewer #2: Yes

5. Review Comments to the Author

Reviewer #1: This was a well designed and presented study. The study was appropriately powered and the statistical analysis was appropriate.

My one concern about the study design was with the Causation and preventability tool, (Table S3). This was the tool used to determine inclusion into the study and was based on subjective assessments of prior hospital care and was scored by the discharging team. Thus this leaves the possibility of underscoring patients who might otherwise meet criteria and thus artificially lowering the percentage of PPR. This should be addressed in the discussion.

Reviewer #2: Dear Authors, it is a very nice design, but I would prefer that this project is either stratified based on specialty or you could split the project into 2 or 3 papers based on the different specialities. Reasons of readmission would be different in surgery from medicine. So putting both in one basket, would affect your results. I would rather put cardiology, gastroenterology, internal medicine, neurology, pulmonology in one group and general surgery in a separate group. Psychiatry also would have different group by itself.

6. PLOS authors have the option to publish the peer review history of their article (what does this mean?). If published, this will include your full peer review and any attached files.

Reviewer #1: Yes: William J Reschly

Reviewer #2: No

---

## [Author Response · Author response to Decision Letter 0]

14 Feb 2020

Response to Editor’s and Reviewer’s comments 

Preventability of unplanned readmissions within 30 days of discharge: A cross-sectional, single-centre study.PONE-D-19-31179

We would like to thank the reviewers for their comments, which helped us to improve the manuscript. We respond on each comment and explain which adjustments were made. 

Kind regards, on behalf of all co-authors,

Albertine van der Does

Amsterdam, 14th of February

Editor’s and Reviewer’s Comments 

 Our Response 

 Location of edits

Journal requirements 

1. Please ensure that your manuscript meets PLOS ONE's style requirements, including those for file naming. The manuscript was checked according to the style requirements, but no discrepancies were found. 

2. Interview guide; For instance, if you developed a guide as part of this study and it is not under a copyright more restrictive than CC-BY, please include a copy, in both the original language and English, as Supporting Information." Two supporting information files were created; the Dutch interview guide (S6 file) and the English translation (S7 file) of that file. Supporting information: S6 file and S7 file

3 a) Please amend your current ethics statement to include the full name of the ethics committee/institutional review board(s) that approved your specific study. The study was approved by the Advisory Committee Scientific Research of OLVG hospital (Advies Commissie Wetenschappelijk Onderzoek). Informed consent was obtained from patients for the interview and to contact the community pharmacist or general practitioner for additional information (e.g., on medication adherence). Line number 103-104 in Methods section

3 b) Once you have amended this statement in the Methods section of the manuscript, please add the same text to the “Ethics Statement” field of the submission form (via “Edit Submission”). done 

4. If you wish to make changes to your Data Availability statement, please describe these changes in your cover letter and we will update your Data Availability statement to reflect the information you provide.

 We do not wish to make changes to our Data Availability statement. 

Reviewer 1 

My one concern about the study design was with the Causation and preventability tool, (Table S3). This was the tool used to determine inclusion into the study and was based on subjective assessments of prior hospital care and was scored by the discharging team. Thus, this leaves the possibility of underscoring patients who might otherwise meet criteria and thus artificially lowering the percentage of PPR. This should be addressed in the discussion. Thank you for this comment. This is indeed correct. We have added this to the strength and weaknesses paragraph: Inclusion was determined by subjective assessment by a member of the discharging department. This could lead to non-inclusion of patients who might have had a PPR, thus preventable readmission cases could therefore have been underestimated. Line number 333-333-335

Reviewer 2 

but I would prefer that this project is either stratified based on specialty or you could split the project into 2 or 3 papers based on the different specialties. Reasons of readmission would be different in surgery from medicine. So, putting both in one basket, would affect your results. I would rather put cardiology, gastroenterology, internal medicine, neurology, pulmonology in one group and general surgery in a separate group. Psychiatry also would have different group by itself.

 We agree that patients from different specialties are not fully comparable. However, the number of (possible preventable) readmissions per specialty was limited (see S8), making it difficult to draw firm conclusions. We wanted to give a broad overview of the preventability of readmissions, but to give insight per specialty we have created S9 illustrating causes of PPR per department. Supporting information: S8 and S9

---

## [Editor Report · Decision Letter 1]

19 Feb 2020

Preventability of unplanned readmissions within 30 days of discharge. A cross-sectional, single-center study.

PONE-D-19-31179R1

Dear Dr. van der Does,

We are pleased to inform you that your manuscript has been judged scientifically suitable for publication and will be formally accepted for publication once it complies with all outstanding technical requirements.

With kind regards,

Peter Dziegielewski, MD, FRCSC

Academic Editor

PLOS ONE
---

## [Editor Report · Acceptance letter]

13 Mar 2020

PONE-D-19-31179R1 

Preventability of unplanned readmissions within 30 days of discharge. A cross-sectional, single-center study. 

Dear Dr. van der Does:

I am pleased to inform you that your manuscript has been deemed suitable for publication in PLOS ONE. Congratulations! Your manuscript is now with our production department. 

With kind regards,

on behalf of

Dr. Peter Dziegielewski 

Academic Editor

PLOS ONE